# Proactive Contact Tracing

**Prateek Gupta**[1,2,3☯]*, **Tegan Maharaj**[1,4☯], **Martin Weiss**[1,4☯], **Nasim Rahaman**[1,5☯], **Hannah Alsdurf**[6‡], **Nanor Minoyan**[7‡], **Soren Harnois-Leblanc**[7‡], **Joanna Merckx**[8‡], **Andrew Williams**[1,4‡], **Victor Schmidt**[1,4], **Pierre-Luc St-Charles**[1], **Akshay Patel**[9], **Yang Zhang**[1], **David L. Buckeridge**[1,8], **Christopher Pal**[1,4], **Bernhard Schölkopf**[5,10], **Yoshua Bengio**[1,4,10]

**1** Montréal Institute of Learning Algorithms (Mila), Montréal, Québec, Canada, **2** The Alan Turing Institute, London, United Kingdom, **3** Department of Engineering Science, University of Oxford, Oxford, United Kingdom, **4** Department of Computer Science and Operations Research, Université de Montréal, Montréal, Québec, Canada, **5** Max Planck Institute for Intelligent Systems, Tübingen, Germany, **6** School of Epidemiology and Public Health, University of Ottawa, Ottawa, Ontario, Canada, **7** Department of Social and Preventive Medicine, School of Public Health, Université de Montréal, Canada, **8** Department of Epidemiology, Biostatistics and Occupational Health, McGill University, Canada, **9** Cheriton School of Computer Science, University of Waterloo, Waterloo, Ontario, Canada, **10** Fellow of the Canadian Institute for Advanced Research (CIFAR), Canada

☯ These authors contributed equally to this work.
‡These authors also contributed equally to this work.
* pgupta@robots.ox.ac.uk

**Data Availability Statement:** Data is collected using the ABM that is coded in Python. Instructions to run the code is available publicly at https://github.com/mila-iqia/COVI-AgentSim (branch: prateek/postplos).

## Abstract

The COVID-19 pandemic has spurred an unprecedented demand for interventions that can reduce disease spread without excessively restricting daily activity, given negative impacts on mental health and economic outcomes. Digital contact tracing (DCT) apps have emerged as a component of the epidemic management toolkit. Existing DCT apps typically recommend quarantine to all digitally-recorded contacts of test-confirmed cases. Over-reliance on testing may, however, impede the effectiveness of such apps, since by the time cases are confirmed through testing, onward transmissions are likely to have occurred. Furthermore, most cases are infectious over a short period; only a subset of their contacts are likely to become infected. These apps do not fully utilize data sources to base their predictions of transmission risk during an encounter, leading to recommendations of quarantine to many uninfected people and associated slowdowns in economic activity. This phenomenon, commonly termed as "pingdemic," may additionally contribute to reduced compliance to public health measures. In this work, we propose a novel DCT framework, Proactive Contact Tracing (PCT), which uses multiple sources of information (e.g. self-reported symptoms, received messages from contacts) to estimate app users' infectiousness histories and provide behavioral recommendations. PCT methods are by design *proactive*, predicting spread before it occurs. We present an interpretable instance of this framework, the *Rule-based PCT* algorithm, designed via a multi-disciplinary collaboration among epidemiologists, computer scientists, and behavior experts. Finally, we develop an agent-based model that allows us to compare different DCT methods and evaluate their performance in negotiating the trade-off between epidemic control and restricting population mobility. Performing extensive sensitivity analysis across user behavior, public health policy, and virological parameters, we compare *Rule-based PCT* to i) binary contact tracing (BCT), which exclusively

**Funding:** Y.B. gratefully acknowledge the following funding sources: NSERC - The Natural Sciences and Engineering Research Council of Canada (RGPIN-2019-04822 | nserccrsng. gc.ca). The funders had no role in study design, data collection and analysis, decision to publish, or preparation of the manuscript.

**Competing interests:** The authors have declared that no competing interests exist.

relies on test results and recommends a fixed-duration quarantine, and ii) household quarantine (HQ). Our results suggest that both BCT and *Rule-based PCT* improve upon HQ, however, *Rule-based PCT* is more efficient at controlling spread of disease than BCT across a range of scenarios. In terms of cost-effectiveness, we show that *Rule-based PCT* pareto-dominates BCT, as demonstrated by a decrease in Disability Adjusted Life Years, as well as Temporary Productivity Loss. Overall, we find that *Rule-based PCT* outperforms existing approaches across a varying range of parameters. By leveraging anonymized infectious-ness estimates received from digitally-recorded contacts, PCT is able to notify potentially infected users earlier than BCT methods and prevent onward transmissions. Our results suggest that PCT-based applications could be a useful tool in managing future epidemics.

## Author summary

The COVID-19 pandemic has overwhelmed the capacity of many governments undertaking contact tracing. Digital tracing applications, which automate the contact tracing process by sensing proximity between users, can limit the spread of infectious diseases, thereby reducing this burden. Though helpful in averting cases, especially when a sufficient number of people use them, such apps, due to the inefficient use of information sources, have important socioeconomic costs when lots of uninfected individuals are asked to stay at home. We proposed a digital contact tracing framework, Proactive Contact Tracing (PCT), which uses multiple sources of information to predict whether a given individual is likely to be infectious on any given day, and recommends appropriately cautious behaviors. We designed *Rule-based PCT* algorithm as an interpretable PCT algorithm, with the rules designed in a close collaboration with epidemiologists, computer scientists, and behavior experts. With the help of a detailed simulator, we examined how *Rule-based PCT* performs in comparison to a) quarantining household members, and b) recommending a fixed quarantine to digital contacts of cases identified through testing. Our cost-effective method was better able to control the epidemic while minimizing restrictions on human activity, under a wide range of simulation parameters. Our PCT framework efficiently leverages data and uses predictions to generate early warning signals and prevent cases from infecting others. Such proactive methods should be considered alongside existing interventions, including in the context of low compliance to social distancing and emergence of highly-infectious variants capable of evading vaccine-based protection.

## Introduction

Governments worldwide have implemented a variety of non-pharmaceutical interventions (NPIs) to contain the spread of COVID-19. Although the strictest of these measures effectively contained case surges in many settings [1], they have had dramatic consequences including long-term impacts on individuals' mental health [2], financial security [3], and on local and global economies [4].

   With global immunization a distant possibility and the emergence of highly infectious variants of concern capable of immune escape [5], public health departments worldwide have, so far, aggressively pursued test-and-trace initiatives to curb viral transmission while relaxing or

removing restrictions on social and economic activities. Traditionally, the tracing process begins each time a COVID-19 case (an 'index case') is reported. For example, in the U.K, public health authorities directly interview index cases to identify any contacts, defined as persons who were recently within two meters of the index case for fifteen minutes or more [6]. This approach, *Manual Contact Tracing*, is time-consuming given the need for follow-up with each individual and challenges in accurately recalling contacts [7, 8]. The scale of the pandemic has consequently stretched the capacity of many health departments undertaking manual tracing [9, 10].

Digital contact tracing (DCT) applications, in contrast, employ electronic devices (e.g., phones using Bluetooth) to record encounters between users, reducing the burden of recall and facilitating notification of contacts. DCT applications can additionally capture characteristics of encounters (e.g., proximity, duration) and anonymously exchange information between users.

Most deployed DCT apps adopt a simple model wherein all digitally-recorded contacts of confirmed index cases receive an app notification recommending quarantine [11]. We refer to such methods as Binary Contact Tracing (BCT) because the inputs consist of binary information about index cases' test results (positive or negative), and the outputs are binary notifications to contacts (quarantine or not). BCT approaches may be sub-optimal for COVID-19 control for the following reasons: First, BCT treats all recorded contacts of an index case as equally infectious; i.e., it does not use contacts' own information (e.g., symptoms, proximity and duration of encounters) to evaluate their likelihood of being infected or infecting others, thereby recommending quarantine to many more people than are infectious [12, 13]. Such overly restrictive notifications can contribute to "pandemic fatigue," leading to reduced compliance [14]. Second, people who never develop symptoms or experience mild symptoms may nevertheless be infectious. A systematic review based on previous SARS-CoV-2 variants suggested that 40% of infections were asymptomatic [15–17], and recent evidence [18] suggests as high as 80–90% of infections with the Omicron variant are asymptomatic. In the absence of population-wide screening and limited testing capacity, such individuals may not qualify for testing. Finally, infectiousness of those who eventually develop symptoms appears to peak around the time of symptom onset [19]. Symptomatic cases are therefore likely to infect others *prior* to being tested, further limiting effectiveness of BCT. Altogether, BCT's exclusive reliance on test results may lead to absent or delayed notification of infectious contacts, undermining epidemic control.

Finally, with the emergence of highly transmissible variants of concern such as Omicron, public health departments have become overwhelmed with test-and-trace approaches. Some nations have even dropped them altogether, shifting responsibility to undertake regular rapid testing and adopt preventive behaviors (including notifying contacts) onto individuals and businesses. Alternative proactive and accurate DCT methods are needed, more than ever, to bridge the gap between BCT and manual tracing methods.

## Proactive Contact Tracing

In this work, we propose *proactive contact tracing* (PCT), a novel contact tracing framework that uses carefully-designed predictors of infectiousness informed by a rich set of features (e.g., pre-existing conditions, daily symptoms, "risk messages") available locally on a given user's smartphone. These predictors privately estimate current and past 14-day infectiousness for that user, enabling *proactive* early warning signals of disease. Discretized values of infectiousness estimates, which we call *risk messages*, are propagated through the network of app users at regular intervals, using a peer-to-peer Bluetooth communication protocol, e.g., COVI [20], or

alternatively, a centralized protocol (see S1 Appendix for a discussion on possible design of such a system). Based on the estimate of the user's current infectiousness, the app recommends one of four sets of increasingly restrictive behaviors, ranging from complete self-isolation to precautions such as avoiding public transport or working from home. A behavior study [21] conducted to understand the efficacy of such an app-based notification system concluded that "less risk-taking by small portions of the population may produce large benefits."

Received risk messages are an input to the local predictor of infectiousness. Therefore, predictions for a given user are indirectly influenced by all other app users. PCT dynamically responds to information as it becomes available (e.g. when informative symptoms are logged), resulting in propagation of an updated risk message at the next interval. Further, PCT improves upon BCT as it allows for less restrictive recommendations in the presence of uncertainty (e.g. in the absence of test results). By suggesting alternative behavioral modifications rather than a fixed-duration quarantine, PCT may impose fewer restrictions on non-infectious individuals, potentially alleviating the economic impact of DCT methods. Hence, PCT not only has the ability to quickly and accurately flag potential infection to a user, but it also proactively generates early-warning signals to recommend cautionary behaviors to likely-to-be-infected users (see Fig 1), thus preventing onward transmission.

PCT relies on designing a predictor that can accurately estimate a user's infectiousness from the available features. This is a non-trivial task because the predictions (and corresponding uncertainties) for one user are fed back into the same predictor for other users through risk messages. Research is underway to develop sophisticated deep-learning-based predictors for this purpose, with preliminary findings suggesting unique design and deployment challenges [22]. For this reason, analyses presented here focus on a simpler, rule-based predictor (described in Methods) which is interpretable to clinicians and public health stakeholders. *Rule-based PCT* is also computationally faster to run, including on legacy smartphones.

## Objectives

Contact tracing methods must negotiate a trade-off between controlling epidemic spread and restricting mobility in a population such that social and economic activities can occur. This study assesses, via a detailed agent-based model (COVI-AgentSim), how well PCT and BCT

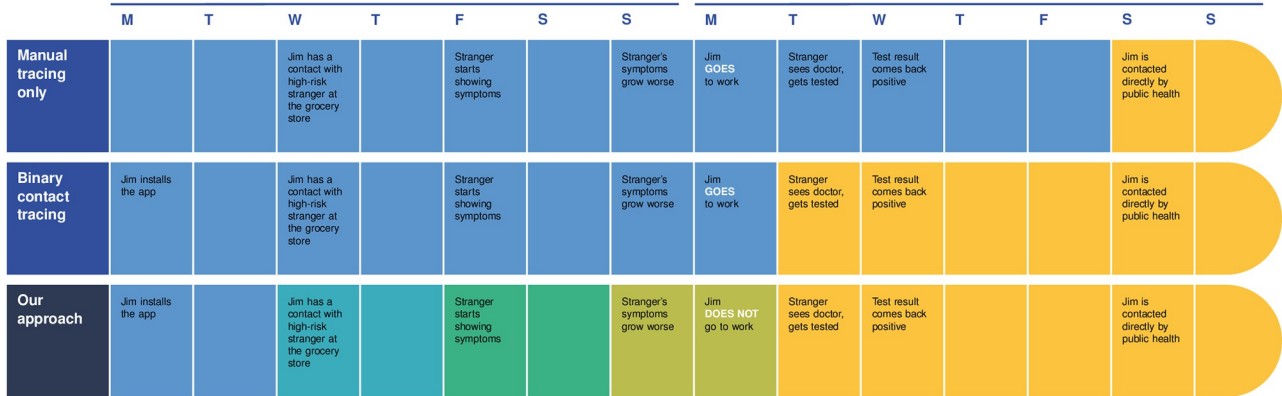

**Fig 1. Motivating example comparing manual, binary, and proactive contact tracing.** This example shows the potential effectiveness of early warning signals in controlling the spread of the infection. We see that manual tracing suffers from a delay between laboratory-confirmed diagnosis and informing all contacts. Further, both manual and digital contact tracing send late signals because they only make use of the strongest possible signal (a laboratory-confirmed diagnosis). The proposed PCT approach takes advantage of reported symptoms and the propagation of risk signals between phones to obtain much earlier signals.

negotiate this trade-off in comparison to *Household Quarantine (HQ)*, which mandates a 14-day quarantine for each confirmed case's entire household. For each method, the app is deployed among a proportion of agents (app users) in a synthetic population. Scenarios estimate the ratio of *cumulative cases* (an indicator of epidemic spread) to *mean number of contacts allowed between agents* (an indicator of population mobility). We further conduct cost-effectiveness analyses across varying degrees of governmental pandemic response. To demonstrate and validate our simulator with real statistics, we focus our analysis on Montreal, Canada (the simulator can be configured to any region for which appropriate statistics are available).

## Methods

To evaluate the performance of contact tracing apps, we proceed as follows: (i) implement an agent-based model to simulate COVID-19 spread in a population of interest, (ii) develop an app-based tracing mechanism which suggests user-behavior modifications, and (iii) compare different contact tracing methods.

### Agent-based model

To simulate the spread of COVID-19 virus in a population and generate naturally-occurring phenomena (e.g. emergence of symptoms) as well as app-based events (e.g. communication of risk messages), we implement COVI-AgentSim (see Code Availability), a discrete event agent-based model that emulates information propagation between two agents as designed in COVI [20], a peer-to-peer communication protocol. Here we present a short description of the simulator; a more detailed description is presented in S2 Appendix.

We sample agent demographics corresponding to the region of Montréal, Quebec, Canada using census data [23] and additional epidemiological data sources. The simulator advances the state of the agents by moving them from one location to another, where locations are either home, workplace/school, or off miscellaneous types. At a specific location $l$, potential encounters are sampled for each agent such that the final aggregated contact pattern resembles that of empirically derived age-stratified contact matrices [24] for the Montréal region. We use the procedure typical of demographic standardization [25] to project the Canada-wide contact matrices to the Montréal region, and further refine this derivation using a scaling factor to accommodate fine-grained information available for the Montréal region [26] (e.g., number of contacts at house or school).

We consider four levels, {1, 2, 3, 4}, of in-app behavior recommendations, with increasing levels restricting the number of close contacts. Whereas an agent in behavior level 0 (not recommended through app) samples the pre-pandemic mean number of contacts per location, behavior level 4 corresponds to quarantine, in which the agent is recommended to stay at their residence, where they do not sample any contacts. Level 3 represents the post-confinement behavior and exhibits a reduced number of contacts compared to the level 0, where the reduction factor follows from empirical surveys [26]. Finally, we introduce two intermediate behavior levels {1, 2} each exhibiting half the reduction factor compared to the next higher level (see Table 1 in S2 Appendix for further details). Our choice of intermediate reduction factors is motivated by simplicity, however, we acknowledge that such reduction factors will likely be influenced by behavior recommendations in these levels, thereby making it crucial to have inputs from PHEs and user behavior experts. As in real life, agents do not always adhere to recommended behavioral restrictions. On a particular day, an agent drops out of their current behavior level to level 1 with a dropout probability [27]. In addition, the simulator assumes imperfect reporting of symptoms or test results by the agents.

Virus transmission, as modeled in the simulator, takes place with a probability [27], $P$, anytime an infectious and a susceptible individual are within 2 meters of each other for at least 15 minutes. When an agent is infected, we sample a proxy measure for viral load, termed Effective Viral Load (EVL), and symptoms to be experienced by the agent for each day of the infection until recovery. EVL for an agent is modeled with a piece-wise linear function varying between 0 and 1, based on parameters such as incubation period, infectiousness onset, and recovery period, sampled according to the published literature [19, 28].

We use age-stratified smartphone/app adoption rates for the population as in [27]. Thus, given a proportion of the population with a smartphone in different age groups, we vary the global uptake parameter to attain a specific adoption rate. The simulator implements a digital communication protocol between agents with smartphones to enable contact tracing methods as discussed next.

## Tracing methods

**Household Quarantine (HQ).**   We consider a baseline scenario in which agents who receive a confirmed positive RT-PCR test are quarantined for 14 days. Other residents of agents' households are also quarantined for the same period.

**Binary Contact Tracing (BCT).**   Under BCT, any agent with an app who reports a positive RT-PCR test result broadcasts this information to alert all of their digitally stored contacts from the past 14 days. Those contacts and their household members are put in level 4, i.e., quarantined, occasionally dropping out as per the dropout probabilities.

**Proactive Contact Tracing (PCT).**   In PCT, the decision about which recommendation level to show to an agent is executed via a predictor. This predictor is part of the contact tracing application on each participating agent's smartphone. The predictor uses the following "features": symptoms, individual characteristics (e.g. age, sex), RT-PCR test results, and anonymized risk messages received from other users (described next) to estimate the agent's mean infectiousness for each of the past 14 days. At regular intervals, each app user's smartphone sends an update to their contacts with a discretized estimate of their infectiousness when the encounter took place; we refer to these as "risk messages". In this study, we focus on a rule-based predictor for the ease of interpretability and computational costs. However, we note that the data generated by Covi-AgentSim could be used to train machine learning algorithms [22].

**Rule-based PCT.**   Rule-based PCT uses a set of heuristics to predict an agent's 14-day risk history. This risk history is an estimate of how likely the agent was infectious over the past 14 days, and can be used to provide behavioural recommendations to the agent and their past contacts. These rules are guided by empirical data and the combined knowledge of experts in relevant domains such as epidemiology, virology and behavior research. Rule-based PCT processes each input feature independently to estimate the agent's infectiousness. For example, the predictor will use the agent's history of symptoms to produce a symptom risk history [29]. After constructing a risk history based on each input feature, the predictor constructs the final risk history by taking the maximum of (a) the highest risk estimated for that day derived from the input features, and (b) the previously estimated risk for that day. As illustration, this method would not recommend to quarantine an agent and their past contacts given only a mild symptom such as a cough, but instead make a more moderate recommendation. Fig 2 shows a simplified flow chart of the algorithm, and S3 Appendix provides the complete algorithm.

## Evaluation

We compare the performance of rule-based PCT, BCT, and HQ scenarios across a range of model parameters that influence user behavior, public health policy, and virological

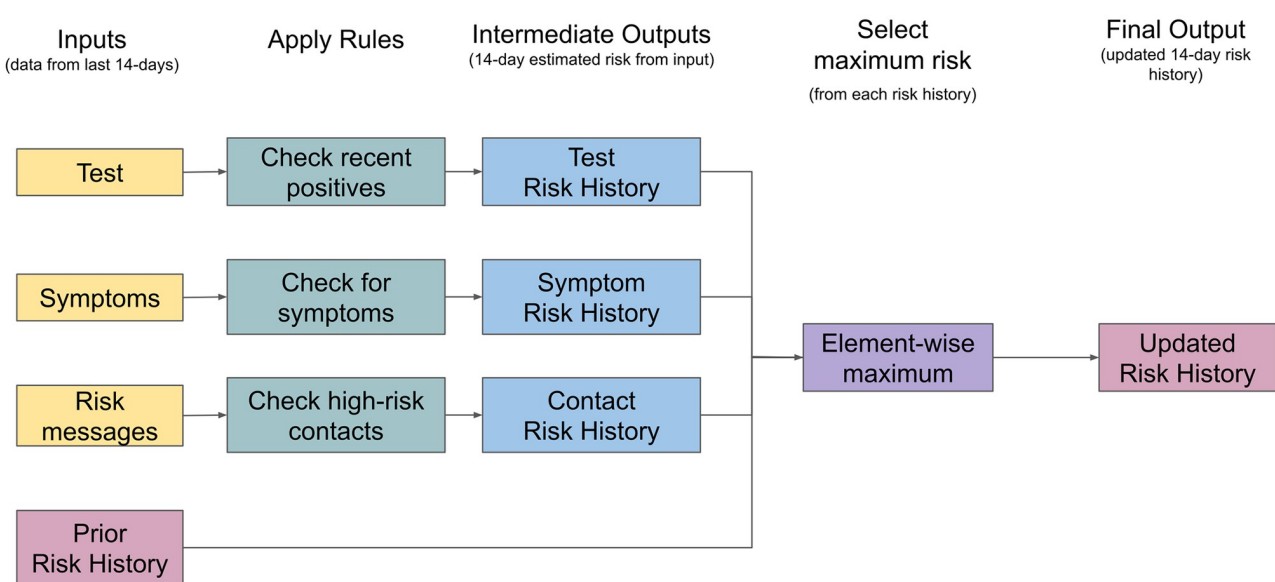

**Fig 2. Rule-Based PCT Overview.** This simplified diagram shows how information flows through the rule-based PCT algorithm. This algorithm is run **each** day on the agent's 14-day history of features to estimate their risk history. Each day the algorithm is run, it takes as input their current RT-PCR test history, user-input symptom history, and anonymized risk message history. Next, a ruleset is applied independently to each input type yielding estimates of the user's risk history over the past 14 days. Finally, to construct a single conservative estimate of the user's risk history, RB-PCT takes the maximum risk across all estimates including the previously generated estimates. There are some exceptions to these rules (e.g. negative test results reset some of the risk history to low values); a full description is provided in S3 Appendix.

assumptions. For each scenario and fixed value of parameters (see Table 1), we aggregate results across 50 randomly seeded, 75-day simulations, each of 10K agents with 40 initial infections. To evaluate the performance of these scenarios across varying degrees of outbreak severity, we run 50 simulations each with a randomly sampled global mobility parameter, $\beta$, defined as the likelihood of foregoing any given contact. For example, under $\beta = 0.6$, *any* sampled contact will be considered a simulated contact with a probability of 0.4. Thus, $\beta$ aims to mimic the strength of government-imposed mobility restrictions such as lockdown and stay-two-meters-

**Table 1. Parameter ranges across which the performance of contact tracing methods is evaluated.**

| Parameter | Description | Minimum value | Maximum value | Default value |
|---|---|---|---|---|
| **User-behavior parameters** | | | | |
| Adoption Rate[†] | The percentage of population using a contact tracing enabled app | 20% | 60% | 60%, 30%* |
| Recommendation Adherence | The percentage of app-users that are likely to follow recommendations on any given day | 36% | 98% | 98% |
| Reporting of symptoms | The likelihood of reporting symptoms for any app-user | 20% | 80% | 80% |
| **Public health policy parameters** | | | | |
| Testing Capacity | Maximum percentage of population that can receive a test on any day | 0.05% | 0.5% | 0.1% |
| **Virological parameters** | | | | |
| Asymptomatic population | The percentage of population that doesn't show any symptoms on getting infected | 20% | 40% | 30% |
| Asymptomatic infectiousness ratio | Infectiousness ratio of asymptomatic cases relative to symptomatic cases | 29% | 45% | 29% |

[†] The parameter is also influenced by public health policy.

*We present all the results across two adoption rates.

apart public campaigns. This parameter allows us to control virus containment under any given scenario by varying the number of daily contacts without changing the virus transmission model.

To compare different scenarios across varying parameter values, we compute the ratio of the cumulative incidence (denoted by $\hat{H}$) and mean daily contacts per agent (denoted by $\hat{E}$), both computed at the end of the simulation. This ratio helps us understand how well a tracing method performs in reducing infection spread while allowing for encounters. We compute $\hat{E}$ by summing up encounters across all the agents throughout the period of the simulation and, finally, dividing by the number of days and the population size to yield mean daily contacts per agent. Thus, $\hat{E}$, roughly, serves as a proxy for economic welfare throughout the simulation. The results presented in the subsequent sections use the mean value of $\frac{\hat{H}}{\hat{E}}$ and 1 standard error to compare the different tracing methods; a lower ratio indicates superior performance.

Finally, to account for aleatoric uncertainty, we bootstrap our simulations to compute model means and standard errors. For each scenario, we draw, with replacement, 1,000 random subsets of size 60 from 60 simulations and compute the mean of the performances for each subset. Thus, we obtain a distribution over the performance metric under each scenario. The plots show means and 1-standard errors of these distributions.

Our simulations are run on a population of 10,000 agents for 75 days. For an efficient PCT algorithm, we expect to recalibrate the Rule-based PCT algorithm based on the evolving attack rates and government restrictions. Thus, for convenience, we chose the specified period of two and a half months to simulate one full epidemic wave. We further evaluate the sensitivity of parameters across 30% and 60% adoption rates. Finally, we show other metrics in S4 Appendix for these simulations.

**Cost-effectiveness analysis.**   To provide additional information on the potential real-life utility of these methods, we perform cost-effectiveness analyses. To quantify the health and economic impacts of different DCT methods, we respectively compute lost Disability-Adjusted Life Years (DALYs) and Temporary Productivity Loss (TPL) for simulations run on COVI-AgentSim. For a reliable cost-benefit analysis, it is necessary to understand the full patient journey and each agent's possible health state until reaching a post-epidemic steady state such that agents that were infected either recovered or died.

Such a trajectory, which is obtained by running the simulations for longer, helps assess whether the contact tracing methods actually avert early death, or simply delay it; miscategorizing the timing and/or death event greatly impacts the overall public health benefits of DCT methods. To account for this challenge, we run longer simulations of 180 days. By the end of these simulations, agents are either susceptible, recovered or dead.

A key challenge decision-makers face is how to allocate resources to maximize public health [30] with the lowest possible economic impact. A tool is needed to compare different health interventions and their associated costs, particularly to weigh the trade-off between improved health outcomes and higher economic costs. The incremental cost-effectiveness ratio (ICER) is a measure that allows for direct comparison across health interventions; the ICER is the difference in costs of two interventions divided by the difference in health outcomes (i.e. effectiveness) of the interventions. The ICER provides critical information for policy makers and enables decision-making on resource allocation and prioritization across healthcare services.

For mathematical definitions of DALYs, TPL, and ICER, we refer the reader to S5 Appendix.

## Results

With the default parameter values, Fig 3 shows, for each method, the fraction of the population that has been infected up to any day, i.e, cumulative incidence (left) and the daily proportion

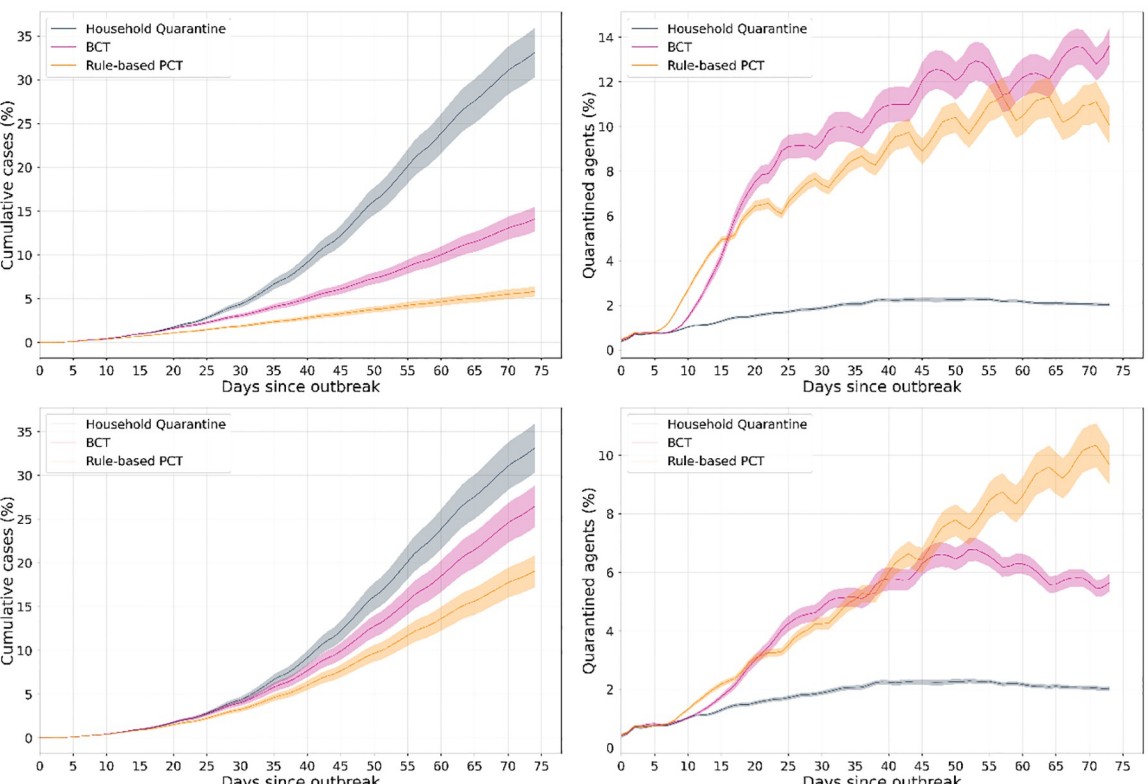

**Fig 3. Left**: Cumulative case counts for each method (fraction of the population) and **Right**: Mobility restriction (fraction of population quarantined), for 60% (**top**) and 30% (**bottom**) app adoption **Gist**: Both BCT and PCT significantly reduce cases as compared to HQ, even at lower app adoption; benefits increase with increasing adoption. With higher adoption, (top) PCT imposes less restriction while achieving much greater reduction in cases. However, at lower adoptions, the reduction in cases is achieved via more conservative recommendations, highlighting the need for more sophisticated predictors if app adoption is low. The plots show mean and 1-standard error bands of these quantities. Note that the HQ scenario includes (false) quarantines because household members of an infected individual are recommended quarantine irrespective of their infection status.

of agents quarantined (right). We observe that PCT results in a lower infected fraction over 60 days at app adoption rates of 60% (top) and 30% (bottom). The fraction of quarantined agents is generally lower for PCT at 60% adoption, indicating that it better negotiates the epidemic control/mobility restriction trade-off. In contrast, the designed rules for PCT did not result in a lower quarantine rate at 30% adoption, suggesting a need for more sophisticated methods for designing these rules [22].

## Adoption rate

App adoption rate is a dominant factor affecting the performance of contact tracing methods. Fig 4 compares the trade-off metric for each method under varying app adoption rates. The observed $\frac{\tilde{H}}{E}$ ratios suggest that PCT improves upon BCT across all adoption rates, even in the context of low uptake (i.e. 20%).

## Sensitivity analyses

We compare how each tracing method performs across the range of parameter values listed in Table 1. We vary one parameter at a time, keeping others at their default values. Fig 5 shows sensitivity to varying the following: (A) Recommendation adherence: proportion of app-users

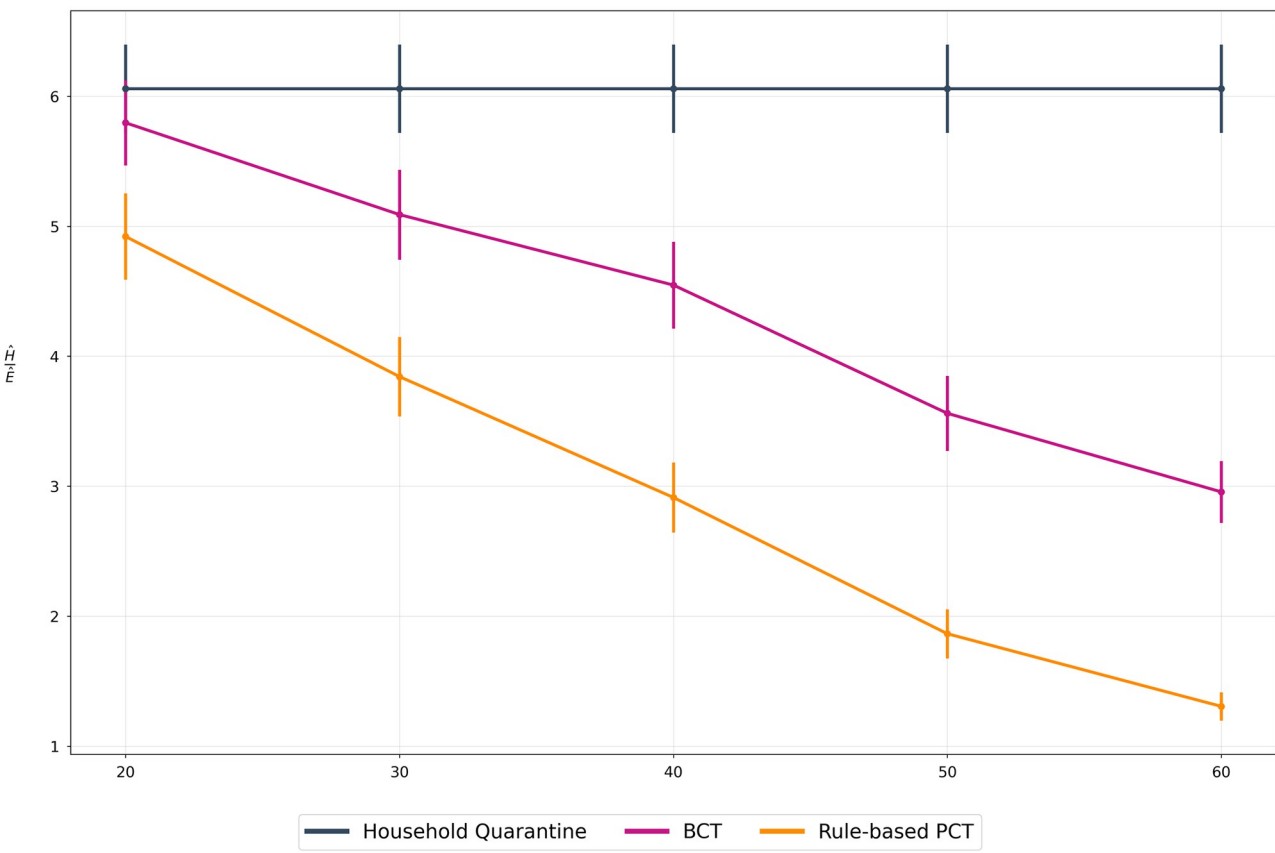

**Fig 4. Adoption rate comparison.** We compare all methods for adoption rates between 0% (HQ) and 60% of both BCT and PCT methods are able to improve over the HQ scenario, even at low adoption rates. We also observe that PCT is able to negotiate the health-economic trade-off better than BCT (lower the ratio, better is the trade-off). We further compare this performance across adoption rates in cost-benefit analysis.

following recommendations on a given day; (B) Symptom reporting: likelihood of daily symptom reporting by app-users; (C) Testing capacity: daily percentage of the population that can receive an RT-PCR test; (D) Asymptomatic fraction: proportion of agents that never develop symptoms. We observe that both BCT and PCT consistently result in lower $\frac{\hat{H}}{\hat{E}}$ ratios compared to HQ, congruent with other studies [27, 31] demonstrating the usefulness of contact tracing apps as components of epidemic management.

Fig 5A suggests that recommendation adherence has a drastic impact on the performance of tracing methods while a modest decrease in performance is found as the probability of reporting symptoms increases (Fig 5B). The relative advantage of PCT over BCT is maintained across a broad range of testing capacities (Fig 5C) and PCT results in a lower $\frac{\hat{H}}{\hat{E}}$ ratio than BCT and HQ at every value of asymptomatic fraction (Fig 5D). At 30% adoption, the trade-off associated with PCT improves as the fraction of asymptomatic cases is increased; this relationship is, however, absent at 60% adoption.

## Infectiousness of asymptomatic cases

Given the emergence of SARS-CoV-2 variants characterized by greater transmission potential, we assess how different methods perform under increased infectiousness of asymptomatic cases relative to symptomatic cases (Fig 6). For all methods, the epidemic control and mobility restriction trade-off worsens as relative infectiousness of asymptomatic cases increases, however, PCT methods remain superior to BCT and HQ.

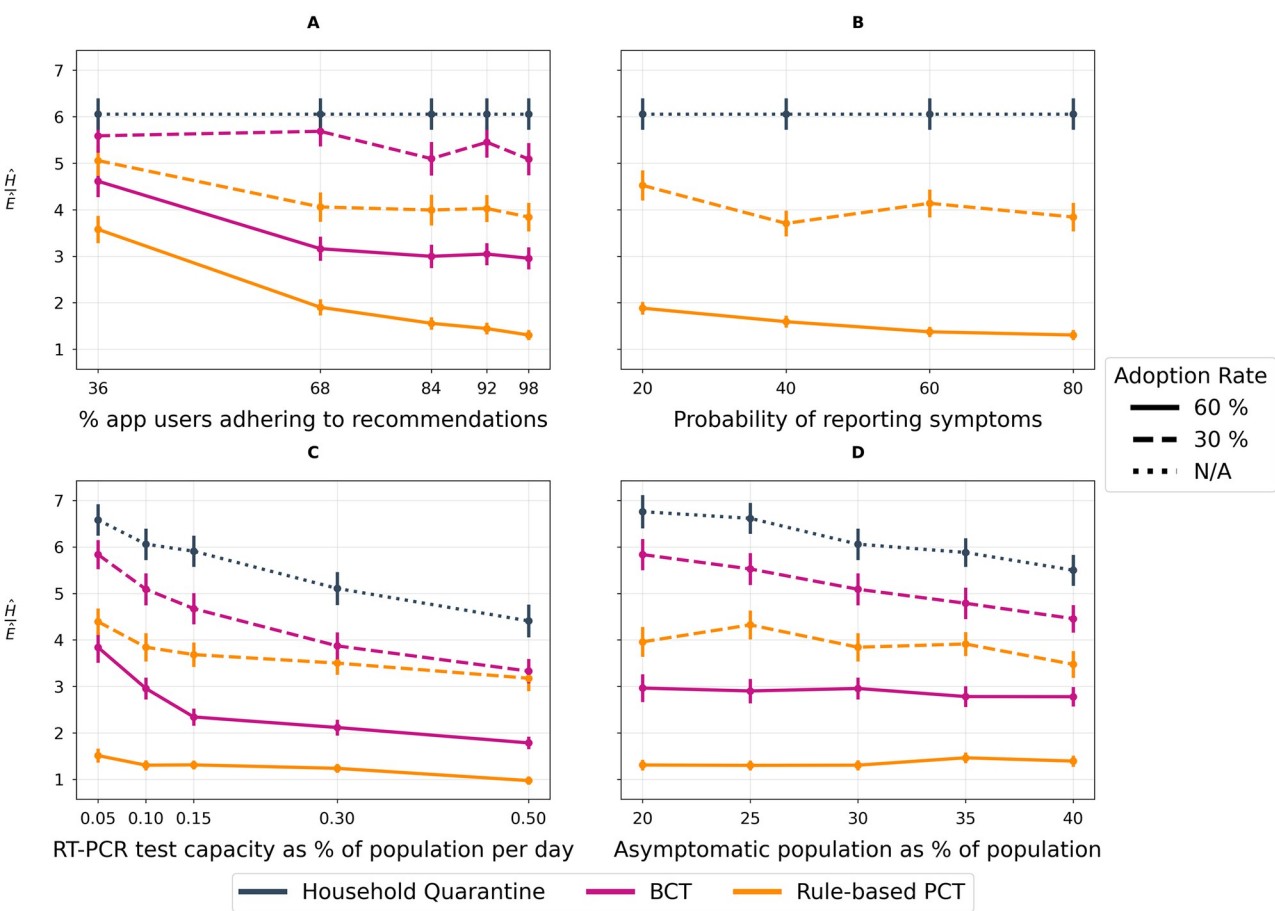

**Fig 5. Sensitivity Analyses.** All experiments measure the proportion of the population infected, $\hat{H}$, within 60 days of an outbreak normalized by mean daily contacts per person per day, $\hat{E}$. We plot $\frac{\hat{H}}{\hat{E}}$ for each tracing method across two app adoption rates as well as against a baseline household quarantining scenario. A lower $\frac{\hat{H}}{\hat{E}}$ ratio indicates a better trade-off between epidemic control and restriction of population mobility. We use N/A to represent irrelevance of adoption rate in the baseline scenario as no DCT app is deployed. **(A) Recommendation Adherence**. Illustrates the impact of varying recommendation adherence (e.g. the daily likelihood of getting a test, quarantining, reducing contacts given an in-app notification is received). **(B) Symptom reporting**. Illustrates the impact of varying the daily rate of symptom reporting. Note: the plot omits BCT because BCT doesn't incorporate symptoms in its inputs. **(C) RT-PCR Testing Capacity**. Illustrates the impact of varying the percentage of the population that can receive an RT-PCR test on any given day, ranging from the observed provincial testing capacity of 0.1% to a highly optimistic value of 0.5% of the population. **(D) Infectiousness and symptoms**. Illustrates the impact of varying the proportion of cases that will not develop symptoms.

## Cost-effectiveness analysis

To better understand the potential societal impact and aid in decision making, Fig 7(a) shows the performance of each scenario across two dimensions quantifying the socio-economic burden of COVID-19: (1) Disability-Adjusted Life Years (DALYs; unit: years) quantifies years of healthy life lost due to disease or poor health [32]; (2) Temporary Productivity Loss (TPL; unit: $) measures the economic cost to society of individuals' temporary absence from work. Under a HQ scenario, with minimal population restrictions, we observe the least TPL but the highest quantity of DALYs, while all DCT scenarios save years of healthy life (lower DALYs) at an economic cost (higher TPL).

To further compare these scenarios pairwise, in Fig 7(b) we calculate incremental cost-effectiveness ratios (ICER), defined as the difference in cost (TPL) between two scenarios divided by the difference in their effect (DALYs). Lower DALYs, TPL, and ICER are desired; see S5 Appendix for a detailed mathematical description of these metrics. Fig 7(b) indicates

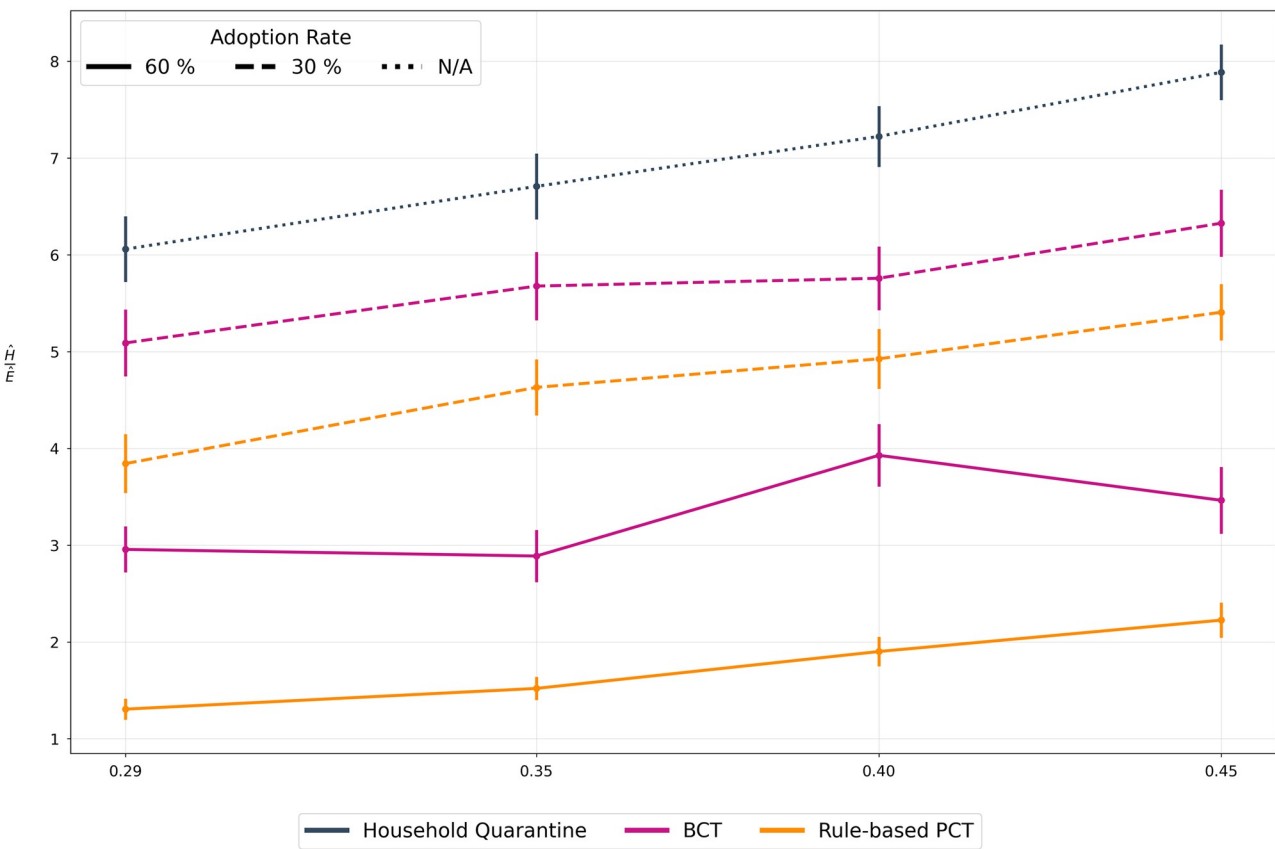

**Fig 6. Asymptomatic infection ratio.** We vary the relative infectiousness of asymptomatic cases. A value of $f$ implies that the asymptomatic case can potentially infect $f$ times as many people as compared to a symptomatic case. A value of 0.29 is the chosen minimum as described in the epidemiological literature while a higher value of 0.45 is a hypothetical situation describing a more infectious variant of the virus. Once again, we use N/A to represent irrelevance of adoption rate in the baseline scenario as no DCT app is deployed. **Gist** As the infectiousness of asymptomatic cases increases, their timely and accurate detection becomes increasingly important. Thus, all the scenarios show a degradation in performance. However, owing to the early warning signals of PCT, it retains its advantage across the range of infection ratios.

that the ICER of PCT is well below that of BCT with HQ as a reference intervention across all adoption rates. Finally, at adoption rates of 30% or more, we observe that PCT is cost-saving with respect to BCT, i.e., PCT results in better health and economic outcomes than BCT, which makes the ICER of PCT with BCT as a reference intervention negative.

## Discussion

### Proactive Contact Tracing

This study presents PCT, a novel digital contact tracing framework. We evaluate how well *Rule-based PCT*, an instantiation of the PCT framework, negotiates the trade-off between epidemic control and restriction of mobility as compared to the existing DCT apps and a baseline household quarantining scenario. Sensitivity analyses conducted across user-behavior, public health policy, and virological parameters show that the *Rule-based PCT* consistently improves over BCT.

Several modeling studies to infer infection status of individuals rely on the partial availability of contact graphs (i.e. who meets whom) which sacrifices user anonymity [34, 35]. By using risk messages, which contain only a few bits (e.g. 2,3,4) at a time, and a well-designed local

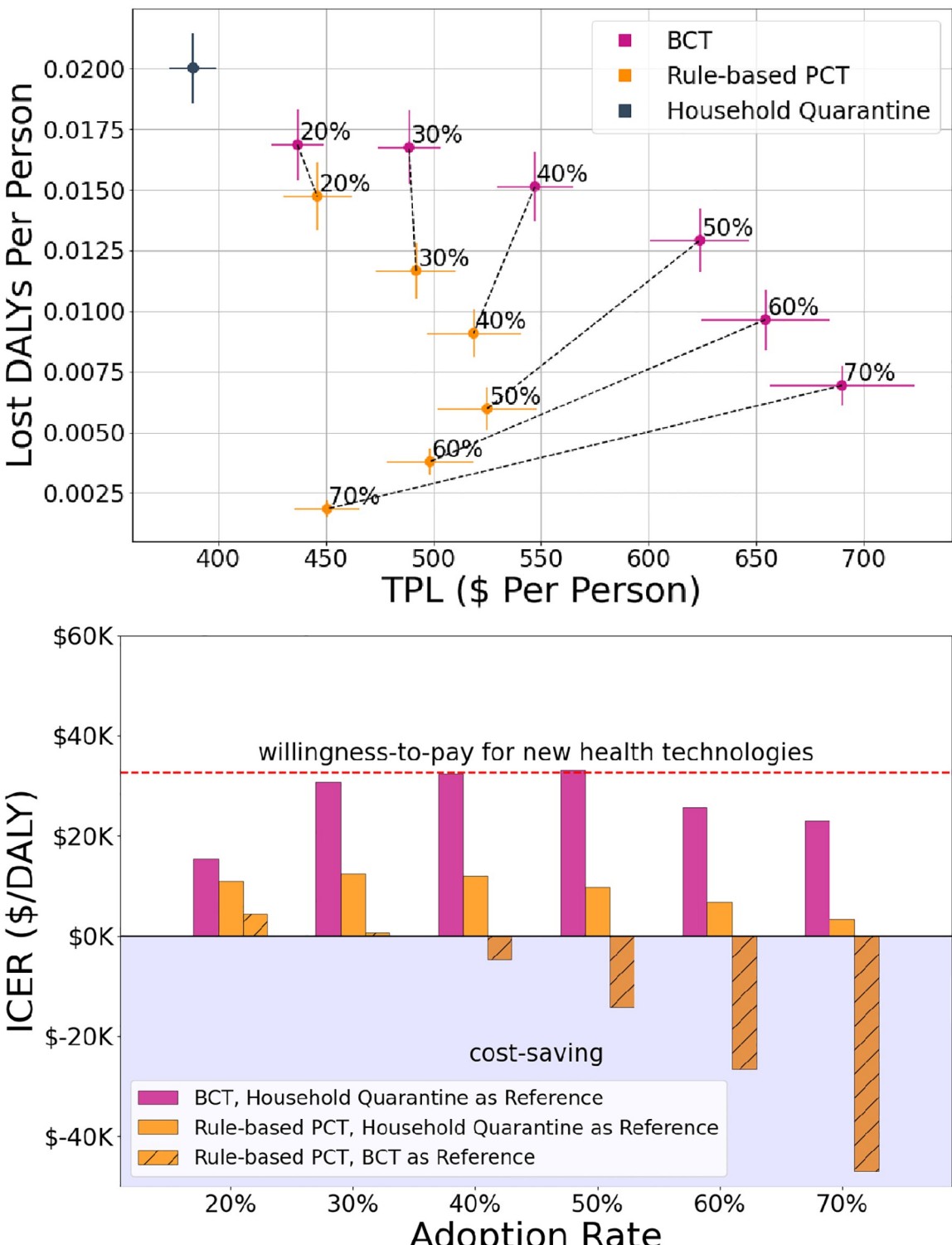

**Fig 7. Cost-effectiveness analysis.** (a) We evaluate the trade-off between temporary productivity loss (TPL) per person and disability-adjusted life years (DALYs) per person for each of the scenarios (HQ, BCT, PCT) at various adoption rates (annotations). (b) Further, we compute incremental cost-effectiveness ratios (ICER) of BCT (pink bars) and PCT (yellow bars) with respect to HQ (unshaded bars) and of PCT with respect to BCT (shaded bars) to quantify these trade-offs with a single metric. The dashed red line represents a willingness-to-pay threshold for new health technologies [33] of $33K in 2020 Canadian dollars (see S5 Appendix for calculations). **Gist** Increased adoption rates lead to better health outcomes for both BCT and PCT, with PCT performing better than BCT across all

adoption rates. Additionally, above 30% adoption, PCT is able to leverage the additional information to dominate BCT, making PCT cost-saving with respect to BCT in this regime. This pareto dominance of PCT over BCT above 30% adoption is shown in figure (b) by negative ICER values of PCT with BCT as a reference intervention (shaded bar). Across all adoption rates, the ICER of PCT is well below the willingness-to-pay threshold for new health technologies. As a final note, increases in the adoption rate of PCT above 50% lead to increasingly cost-saving outcomes.

predictor, PCT is able to accurately identify infections while preserving user-privacy. To account for uncertainty in predictions, PCT provides personalized cautionary recommendations reducing users' mobility instead of completely halting their activity.

COVI-AgentSim [36] was designed with the possibility of generating training data for machine learning algorithms [22]. While this latter approach holds great promise, a significant advantage of rule-based PCT remains its interpretability. These rules can be further refined to reflect evolving knowledge and regional particularities.

### Strengths

Previous studies suggest that effectiveness of DCT apps is strongly correlated with their level of adoption [27, 37], but significant benefits are nevertheless observable at low adoption [35, 38]. In our simulations, PCT consistently fares better than BCT in negotiating the epidemic control/mobility restriction trade-off across the range of adoption rates.

Under limited RT-PCR testing, we expect asymptomatic cases to mainly be identified through contact tracing. Sensitivity analyses thus examined the impact of varying the asymptomatic fraction, as well as the infectiousness of asymptomatic agents. Fig 5D suggests that an increase in the number of asymptomatic agents in the population requires greater precision to control the epidemic while allowing for encounters to take place. Fig 6 shows that increasing the contagiousness of asymptomatic agents results in reduced performance of DCT. Importantly, in the PCT framework, risk messages provide preliminary clues about potentially infected cases even in the absence of test results, thereby generating early warning signals to rapidly caution infectious individuals, and breaking the chain of infections. Both figures show that PCT consistently exhibits a superior trade-off compared to BCT, which relies on testing.

A key advantage of PCT is its ability to integrate individualized data (i.e. symptoms, testing, etc.) in a cost-effective and privacy-preserving manner. Our cost-effectiveness analyses demonstrate that even at low adoption rates, PCT yields better health outcomes and a lower ICER with respect to HQ than BCT. The ICER of PCT with respect to HQ is well below standard willingness-to-pay thresholds (the Canadian threshold adjusted for inflation for new health technologies is $33K per DALY averted in 2020 Canadian dollars [33]; see S5 Appendix for calculations). Furthermore, for higher adoption rates ($\geq 40\%$), PCT is actually cost-saving compared to BCT. As the adoption rate increases, PCT results in both better health *and* economic outcomes, which leads to the reversal observed in Fig 7(a). This reversal corresponds to the decreasing ICERs observed in Fig 7(b). PCT dominates BCT by leveraging additional information, and thus overcomes a trade-off often perceived as unavoidable.

### Limitations

The level of detail in our agent-based model allows more realistic individual-level simulation, however the added complexity may also introduce limitations. For example, epidemiological parameters like asymptomatic fraction are inherently difficult to measure, and may change with new variants of the virus, notably, the Omicron strain, which our study predates. Nevertheless, sensitivity analyses varied parameters we judged as having highly uncertain values. While the ordering in performance of methods remained invariant to these parameters, other

sources of bias and uncertainty may be important and should be examined prior to real-world trials.

We opted for a univariate sensitivity analysis for simplicity and because multi-variate analysis would be computationally expensive. Furthermore, multi-variate analysis would require input from decision-makers to stress-test appropriate scenarios. Decision-makers might also want to attribute non-equal weights to incidence and mean daily contacts in calculating $\frac{\hat{H}}{\hat{E}}$, given varying health and economic priorities.

We emphasize that PCT has been designed to leverage the agility and scalability of DCT apps while integrating estimation of infectiousness, which is implicitly performed by health agents during manual contact tracing. As a result, we do not consider DCT apps to replace manual tracing; the two approaches can be used complementarily, and their interaction would be an interesting avenue for future study [39]. In addition, as in [27, 40], we indirectly accounted for complementary policies (e.g., wearing mask, distancing, etc.) by varying the global mobility factor (see Methods).

Finally, cost-effectiveness analyses used an aggregated TPL calculation, as financial data was not available to allow for a micro-costing approach. Although this may fail to account for how costs vary by context, temporary decreases in productivity at the societal level may still be a concern for decision-makers. To establish the acceptability of our proposal, we used a willingness-to-pay threshold adjusted for inflation according to the Canadian CPI. These thresholds are not standardized, i.e., they vary across the globe as well as the nature of intervention (e.g., technology and vaccine will have a different threshold). Thus, for a more detailed acceptability analysis we think it will be useful to establish these thresholds across important categories.

## Next steps

As mentioned, more sophisticated deep learning (DL) predictors trained on the output of COVI-AgentSim show great promise, particularly to overcome issues relating to low app adoption. Improving interpretability of DL methods, and using them to inform future iterations of the rule-based predictor, constitute an important next step. This direction will not only contribute to efforts in responsible AI deployment, but also lead to crucial methodological developments. Moreover, once the PCT app is deployed, incorporating real-world app usage data with the simulated data can be used to learn a more accurate generative model (simulator). This can improve understanding of infection propagation mechanisms across communities, further allowing tailored and efficient responses.

We acknowledge that app adoption can vary among regions or communities, which might induce disproportionate impacts on health and economic outcomes. Both *Rule-based PCT* and any deep learning method can be tailored to reflect such inequities in adoption through inference from time-limited anonymised app data. This direction of research, though challenging, will be necessary to ensure equal distribution of public goods.

Finally, we note that COVID-19 is an overdispersed epidemic [41], whereby a few people infect many while most people infect a few. Identifying the source of infection, which is the focus of *backward tracing*, may favour tracing efficiency. The PCT framework can support backward tracing by reserving bits of information in risk messages to indicate the likelihood of contagion; we aim to design such a predictor in future work.

To conclude, we presented a PCT framework to improve upon existing DCT framework. We further designed *Rule-based PCT* algorithm as one instantiation of the PCT framework, and compared its performance against the existing DCT approach. Given the speed and scale at which apps can be deployed as compared to vaccine development and testing, proactive

contact tracing apps may prove effective during emerging epidemics and present a valuable complementary strategy to existing NPIs in the presence of vaccine-evading variants.

## Supporting information

**S1 Appendix. Centralized and Decentralized versions of PCT application.** Discussion on possible designs of PCT based applications.
(PDF)

**S2 Appendix. Agent-based model.** Further details of COVI-AgentSim.
(PDF)

**S3 Appendix. Rule-based PCT.** Algorithmic description of Rule-based PCT.
(PDF)

**S4 Appendix. Other simulation metrics.**
(PDF)

**S5 Appendix. Cost-benefit analysis.** More details on DALYs, TPL, and ICER.
(PDF)

## Acknowledgments

We thank all members of the broader COVI project (https://mila.quebec/en/project/covi/) for their dedication and teamwork; it was a pleasure and honour to work with such a great team. This project could not have been completed without the resources of Compute Canada and Calcul Quebec, in particular the Beluga cluster. PG extends his thanks to Tim Ecott and Prateek Bansal for their support and helpful comments on the draft.

## Author Contributions

**Conceptualization:** Prateek Gupta, Tegan Maharaj, Martin Weiss, Nasim Rahaman, Hannah Alsdurf, Nanor Minoyan, Soren Harnois-Leblanc, Joanna Merckx, Andrew Williams, Victor Schmidt, David L. Buckeridge, Yoshua Bengio.

**Data curation:** Prateek Gupta, Martin Weiss, Nasim Rahaman, Nanor Minoyan, Andrew Williams, Victor Schmidt, Pierre-Luc St-Charles, Akshay Patel, Yang Zhang.

**Formal analysis:** Prateek Gupta, Tegan Maharaj, Martin Weiss, Nasim Rahaman, Hannah Alsdurf, Nanor Minoyan, Soren Harnois-Leblanc, Joanna Merckx, Andrew Williams, Pierre-Luc St-Charles, Akshay Patel, Yang Zhang, David L. Buckeridge, Christopher Pal, Bernhard Schölkopf, Yoshua Bengio.

**Funding acquisition:** Yoshua Bengio.

**Investigation:** Prateek Gupta, Tegan Maharaj, Martin Weiss, Nasim Rahaman, Hannah Alsdurf, Nanor Minoyan, Soren Harnois-Leblanc, Joanna Merckx, Andrew Williams, Victor Schmidt, Pierre-Luc St-Charles, Akshay Patel, Yang Zhang, David L. Buckeridge, Christopher Pal, Bernhard Schölkopf, Yoshua Bengio.

**Methodology:** Prateek Gupta, Tegan Maharaj, Martin Weiss, Nasim Rahaman, Hannah Alsdurf, Nanor Minoyan, Soren Harnois-Leblanc, Joanna Merckx, Andrew Williams, Victor Schmidt, Pierre-Luc St-Charles, Yang Zhang, David L. Buckeridge, Bernhard Schölkopf, Yoshua Bengio.

**Project administration:** Prateek Gupta, Tegan Maharaj, Yoshua Bengio.

**Resources:** Prateek Gupta, Bernhard Schölkopf.

**Software:** Prateek Gupta, Tegan Maharaj, Martin Weiss, Nasim Rahaman, Andrew Williams, Victor Schmidt, Pierre-Luc St-Charles, Akshay Patel.

**Supervision:** Prateek Gupta, Yang Zhang, David L. Buckeridge, Christopher Pal, Bernhard Schölkopf, Yoshua Bengio.

**Validation:** Prateek Gupta, Tegan Maharaj, Martin Weiss, Nasim Rahaman, Hannah Alsdurf, Soren Harnois-Leblanc, Joanna Merckx, Andrew Williams, Victor Schmidt, Akshay Patel, Yang Zhang, David L. Buckeridge, Bernhard Schölkopf, Yoshua Bengio.

**Visualization:** Prateek Gupta, Martin Weiss, Hannah Alsdurf, Soren Harnois-Leblanc, Joanna Merckx, Andrew Williams, Yang Zhang.

**Writing – original draft:** Prateek Gupta, Tegan Maharaj, Martin Weiss, Nanor Minoyan, Soren Harnois-Leblanc, Andrew Williams.

**Writing – review & editing:** Prateek Gupta, Tegan Maharaj, Martin Weiss, Nasim Rahaman, Hannah Alsdurf, Nanor Minoyan, Soren Harnois-Leblanc, Joanna Merckx, Andrew Williams, Victor Schmidt, Pierre-Luc St-Charles, Akshay Patel, Yang Zhang, David L. Buckeridge, Christopher Pal, Bernhard Schölkopf, Yoshua Bengio.

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
