## [Decision Letter · Decision Letter 0]

1 Nov 2022

PDIG-D-22-00143

Proactive Contact Tracing

PLOS Digital Health

Dear Dr. Gupta,

Thank you for submitting your manuscript to PLOS Digital Health. After careful consideration, we feel that it has merit but does not fully meet PLOS Digital Health's publication criteria as it currently stands. Therefore, we invite you to submit a revised version of the manuscript that addresses the points raised during the review process.

Please address Reviewer 2's questions and comments in a revised version. 

Please submit your revised manuscript within 60 days Dec 31 2022 11:59PM. If you will need more time than this to complete your revisions, please reply to this message or contact the journal office at digitalhealth@plos.org. Please include the following items when submitting your revised manuscript:

We look forward to receiving your revised manuscript.

Kind regards,

Yuan Lai, Ph.D.

Academic Editor

PLOS Digital Health

Journal Requirements:

1. Please ensure that you provide a single, cohesive .tex source file for your LaTeX revision. You may upload this file as the item type 'LaTeX Source File.' As stated in the PLOS template, your references should be included in your .tex file (not submitted separately as .bib or .bbl). Please also ensure that you are making any formatting changes to both your .tex file and the PDF of your manuscript. If you have any questions, please contact Latex@plos.org. You can find our LaTeX guidelines here: https://journals.plos.org/digitalhealth/s/latex

2. Please update your Author Summary. The aim should be to make your findings accessible to a wide audience that includes both scientists and non-scientists. Sample summaries can be found on our website under Submission Guidelines: https://journals.plos.org/digitalhealth/s/submission-guidelines#loc-parts-of-a-submission.

3. Please provide separate figure files in .tif or .eps format and ensure that all files are under our size limit of 10MB. If you are using LaTeX, you do not need to remove embedded figures.

For more information about how to convert your figure files please see our guidelines: https://journals.plos.org/digitalhealth/s/figures

Additional Editor Comments (if provided):

Reviewers' comments:

Reviewer's Responses to Questions

**Comments to the Author**

1. Does this manuscript meet PLOS Digital Health’s publication criteria? Is the manuscript technically sound, and do the data support the conclusions? The manuscript must describe methodologically and ethically rigorous research with conclusions that are appropriately drawn based on the data presented.

Reviewer #1: Yes

Reviewer #2: Partly

2. Has the statistical analysis been performed appropriately and rigorously?

Reviewer #1: Yes

Reviewer #2: Yes

3. Have the authors made all data underlying the findings in their manuscript fully available (please refer to the Data Availability Statement at the start of the manuscript PDF file)?

Reviewer #1: Yes

Reviewer #2: Yes

4. Is the manuscript presented in an intelligible fashion and written in standard English?

Reviewer #1: Yes

Reviewer #2: Yes

5. Review Comments to the Author

Reviewer #1: Various contact tracing methods have been proposed and implemented, but as the authors pointed out, they have had a significant economic impact, for example, leading to excessive isolation. In this study, the authors showed that more effective measures can be taken by using not only contact tracing but also information on the background of the case.

Although the study is based on simulation and there are significant barriers to actual implementation, it is interesting as a study.

The results of the peer review have been clearly answered and I believe that this manuscript can be adopted.

Reviewer #2: The authors have proposed a protocol for digital contact tracing and used a simulation to compare the novel protocol with home quarantine and binary (digital) contact tracing. 

The authors have addressed the concerns of previous reviewers sufficiently, with the exception of the issue of DCT system architecture and privacy raised by Reviewer #2. 

I'm unable to visualize how PCT would be implemented in a centralized or decentralized DCT architecture. 

So, I don't understand:

1. how the peer-to-peer mechanism is not critical to results (as stated in the response to reviewer #2) when peer-to-peer communication seems to be the only way the risk messages would be received in a decentralized DCT system. 

2. how the current and past 14 days infectiousness is estimated privately (line 55)

3. how the protocol integrates individualized data in a privacy-preserving manner (line 344)

An explanation to suggest how the rule based PCT protocol could be implemented in decentralized and centralized 

architectures, perhaps in an appendix, would provide clarity to the authors' statements about centralized and decentralized architectures and privacy. A diagram of the data flow and architecture would be quite useful. 

With regard to the cost effectiveness analysis, the 20K Canadian dollar per DALY averted threshold is out of date. The article referenced is from 1992 and uses 1990 Canadian dollars. The current manuscript doesn't include a PPP adjustment, so it's likely to be an underestimate of current willingness to pay thresholds. The analysis should use a more contemporary threshold for Canada. This will affect the comparison of the ICER of BCT and PCT with the willingness to pay threshold.

6. PLOS authors have the option to publish the peer review history of their article (what does this mean?). If published, this will include your full peer review and any attached files.

**Do you want your identity to be public for this peer review?** For information about this choice, including consent withdrawal, please see our Privacy Policy.

Reviewer #1: No

Reviewer #2: No

---

## [Decision Letter · Decision Letter 1]

25 Jan 2023

Proactive Contact Tracing

PDIG-D-22-00143R1

Dear Mr. Gupta,

We are pleased to inform you that your manuscript 'Proactive Contact Tracing' has been provisionally accepted for publication in PLOS Digital Health.

Best regards,

Yuan Lai, Ph.D.

Academic Editor

PLOS Digital Health

Reviewer Comments (if any, and for reference):

Reviewer's Responses to Questions

**Comments to the Author**

1. If the authors have adequately addressed your comments raised in a previous round of review and you feel that this manuscript is now acceptable for publication, you may indicate that here to bypass the “Comments to the Author” section, enter your conflict of interest statement in the “Confidential to Editor” section, and submit your "Accept" recommendation.

Reviewer #2: All comments have been addressed

2. Does this manuscript meet PLOS Digital Health’s publication criteria? Is the manuscript technically sound, and do the data support the conclusions? The manuscript must describe methodologically and ethically rigorous research with conclusions that are appropriately drawn based on the data presented.

Reviewer #2: Yes

3. Has the statistical analysis been performed appropriately and rigorously?

Reviewer #2: Yes

4. Have the authors made all data underlying the findings in their manuscript fully available (please refer to the Data Availability Statement at the start of the manuscript PDF file)?

Reviewer #2: Yes

5. Is the manuscript presented in an intelligible fashion and written in standard English?

Reviewer #2: Yes

6. Review Comments to the Author

Reviewer #2: Thank you for addressing my concerns. The protocols, data flows and effects of system architecture are clearer to me now.

I agree that Laupacis et al. (1992) provides the best available WTP threshold given that it pertains to technology adoption instead of oncology or some other medical field. The adjustment for inflation makes the threshold better but I don't expect it to account for changes in attitudes towards technology given the extent of digitization over the last 30 years.

7. PLOS authors have the option to publish the peer review history of their article (what does this mean?). If published, this will include your full peer review and any attached files.

**Do you want your identity to be public for this peer review?** For information about this choice, including consent withdrawal, please see our Privacy Policy.

Reviewer #2: No
